# Sensory Gating during Voluntary Finger Movement in Amyotrophic Lateral Sclerosis with Sensory Cortex Hyperexcitability

**DOI:** 10.3390/brainsci13091325

**Published:** 2023-09-14

**Authors:** Toshio Shimizu, Yuki Nakayama, Kota Bokuda, Kazushi Takahashi

**Affiliations:** 1Department of Neurology, Tokyo Metropolitan Neurological Hospital, Tokyo 183-0042, Japan; kouta_bokuda@tmhp.jp (K.B.); kazushi_takahashi@tmhp.jp (K.T.); 2Unit for Intractable Disease Nursing Care, Tokyo Metropolitan Institute of Medical Science, Tokyo 156-8506, Japan; nakayama-yk@igakuken.or.jp

**Keywords:** amyotrophic lateral sclerosis, somatosensory evoked potential, sensory cortex, hyperexcitability, sensory gating, frontal N30

## Abstract

Cortical responses in somatosensory evoked potentials (SEP) are enhanced in patients with amyotrophic lateral sclerosis (ALS). This study investigated whether sensory gating is involved in the pathophysiology of sensory cortical hyperactivity in ALS patients. The median nerve SEP was recorded at rest and during voluntary finger movements in 14 ALS patients and 13 healthy control subjects. The parietal N20, P25, and frontal N30 were analyzed, and sensory gating was assessed by measuring the amplitude of each component during finger movement. The amplitudes of the N20 onset–peak, N20 peak–P25 peak, and N30 onset–peak were higher in ALS patients than in controls. Nonetheless, there were no significant differences in the amplitude reduction ratio of SEPs between patients and controls. There was a significant correlation between the baseline amplitudes of the N20 onset–peak or N20 peak–P25 peak and their gating ratios in patients with ALS. Our findings indicate that the excitability of the primary sensory cortex and secondary motor cortex is enhanced in ALS, while sensory gating is preserved in the early stages of ALS. This result suggests that enhanced SEP is caused by the hyperexcitability of the primary sensory and secondary motor cortices but not by the dysfunction of inhibitory mechanisms during voluntary movements.

## 1. Introduction

Amyotrophic lateral sclerosis (ALS) is a representative motor neuron disease in adults. Accumulated evidence, however, indicates that ALS is multisystem neuronal degeneration including the peripheral and central sensory pathways and sensory cortices [1,2,3]. Recent studies on the median nerve somatosensory evoked potentials (SEPs) reported that the early cortical components of the SEPs and their high-frequency oscillations (HFOs) were enlarged in patients with ALS [4,5,6], and that a high N20–P25 amplitude was an independent prognostic factor for short survival in ALS [6].

The enlargement of the N20 component may be due to three factors. First, the excitability change of the primary motor cortex may lead to compensatory or neuroplastic changes in the sensory cortex [4]. Paired-pulse transcranial magnetic stimulation has shown that motor cortical hyperexcitability occurs in the early stage of ALS [7,8] and may secondarily induce alterations in the excitability of the primary sensory cortex. Second, early ALS-specific neuropathological changes induced by the subclinical accumulation of phosphorylated TAR DNA-binding protein-43 (TDP-43) may be present in the sensory cortex [1,2]. Third, sensory gating function at the spinal and cortical levels may be impaired in ALS. Sensory gating can originate in different central sensory neurons, including the spinal cord, brainstem, thalamus, and neocortex, where top-down (corticofugal) and bottom-up (corticopetal) inhibitory functions play a pivotal role [9,10,11,12].

This study investigated whether sensory gating function is impaired in patients with ALS to elucidate a pathophysiology of enlarged cortical components of the median nerve SEP in ALS.

## 2. Materials and Methods

### 2.1. Participants

Fourteen patients (nine males and five females) with sporadic ALS participated in the study. ALS was classified as clinically definite (*n* = 3), clinically probable (*n* = 4), or clinically possible (*n* = 4) ALS according to the Awaji criteria [13]. We also included three patients with progressive muscular atrophy (PMA) because PMA shares a common pathophysiology with ALS [14]. PMA has been determined as a distinct phenotype of ALS, and neuropathological studies have shown subclinical lesions in the pyramidal tract in patients with PMA [15,16]. The new diagnostic criteria of ALS (so-called Gold Coast criteria) include PMA [17]. All of the patients fulfilled the Gold Coast criteria. Patient profiles are shown in Table 1. All patients had a relentlessly progressive course. At the time of examination, all patients were ambulatory with or without assistance or wheelchair bound, and did not complain of respiratory discomfort, although their respiratory function was variable. The onset region, disease duration, total score on the Amyotrophic Lateral Sclerosis Functional Rating Scale-Revised (ALSFRS-R) [18], body mass index, and forced vital capacity at the time of examination are shown in Table 1. We did not perform ALS-related gene analyses for all patients in this study.

Thirteen age- and sex-matched healthy control subjects (seven males and six females) served as controls (Table 1).

### 2.2. Nerve Conduction Study

Compound muscle action potential (CMAP) and sensory nerve action potential (SNAP) were also measured by electrical stimulation of the median nerve at the wrist in the more paretic hand in patients and in the left hand in control subjects. CMAP was recorded from the abductor pollicis brevis muscle using surface Ag/AgCl electrodes placed over the muscle belly and tendon. The stimulus intensity was supramaximal with a 0.2 ms square wave current. SNAP was antidromically recorded 20 times from the index finger using ring electrodes placed at the proximal and distal interphalangeal joints of the index finger. The baseline-to-peak amplitudes of CMAP and SNAP were measured. The filter setting was set at 5 Hz to 5 kHz. The skin temperature of the hand was maintained at ≥32 °C.

### 2.3. Somatosensory Evoked Potential

Median nerve SEPs were recorded using AgCl electrodes (8 mm in diameter), which were glued to the scalp sites, neck, and Erb’s point. Electrodes were placed on the parietal region 2 cm posterior to C3 or C4 (CP) and on the frontal region (F3 or F4) contralateral to the stimulation side. The reference electrode was placed on Fz to record parietal components (N20 and P25) and was attached to both ear lobes as an extracephalic reference for the recording of frontal components (N30). Ear lobe potentials were averaged (A+). Electrode impedance was maintained at <5 kΩ. A ring electrode was attached to the index finger of the stimulated side, and SNAPs were simultaneously monitored during median nerve stimulation.

The participants were placed supine on a bed in a quiet, air-conditioned, and electrically shielded room and were asked to be awake during the recording. The hand was placed in a prone position so that each finger could be moved without producing undue stimulus-site displacements over the wrist. The median nerve was electrically stimulated at the wrist, and each median nerve was examined separately in patients and controls. The duration of the electrical stimulus was 0.2 ms, and the stimulation frequency was 5 Hz. The stimulus intensity was set to a level that evoked a slight movement of the thumb. A total of 500 samples were averaged using the Neuropack MEB series (Nihon-Kohden, Tokyo, Japan) for each recording session. The filter setting was 1 to 3000 Hz, and the sensitivity and sweep time were 20 mV and 5 µs per division, respectively. The analysis time was 100 ms after the stimulation.

First, the SEP was recorded at baseline at rest in the hand and fingers of the stimulation side. Particular care was taken to ensure that the hand was completely relaxed and the fingers were immobile. At least two sets of measurements were made at baseline to confirm the reproducibility of results. Then, the participant was asked to move each finger of the stimulated hand alternatively to produce the act of “playing the piano” at an approximate rate of one to two movements per second (task; Figure 1) [19,20]. The pacing rhythm of the movements was left at the discretion of each participant, without any pacing sound. The participants were asked to move their fingers in such a way as to avoid rubbing the fingers against each other. The patients with muscle weakness were asked to move their fingers as much as possible. All the participants were able to perform finger movements during the recording sessions. Two sets of measurements were made in each task session.

All SEP components were recorded in the hemisphere contralateral to the stimulation. The following SEP parameters were measured: the onset-to-peak amplitude of N20 (N20o–p) and peak-to-peak amplitude between N20 and P25 (N20p–P25p) for parietal components (CP3 or CP4 with Fz reference), and peak-to-peak amplitude between P22 and N30 (N30o–p) for frontal components (F3 or F4 with A+ reference). The amplitudes of two-set recordings measured at baseline and task session were averaged after the examinations in each participant. The gating ratio for each SEP (N20o–p, N20p–P25p, and N30o–p) was calculated as the amplitude ratio in the task session relative to the baseline.

### 2.4. Statistical Analysis

The results are reported as mean ± SD. Significant differences between the two groups were evaluated using Welch’s *t*-test. The SEP values recorded from both hemispheres were analyzed together in each group; each participant had two datasets of each hemisphere for both at rest and during the task. The correlations between the baseline amplitude and gating ratio for N20o–p, N20p–P25p, and N30o–p in ALS patients and controls were examined using Pearson’s correlation coefficient. Statistical analyses were performed with JMP version 13.0.0 (SAS Institute, Cary, NC, USA) for Macintosh. All tests were two-sided, and *p*-values of less than 0.05 were considered significant.

## 3. Results

The CMAP amplitude was significantly smaller in patients than in control subjects (Table 1). The SNAP amplitude did not differ significantly between these groups (Table 1).

The amplitudes of SEP components were significantly higher in patients for all of N20o–p, N20p–P25p, and N30o–p than those in control subjects (Table 2, Figure 2 and Figure 3A). The results in the three PMA patients (data from six median nerves) were also significantly higher than those in the control subjects (Table 2).

The amplitude ratios in patients showed no significant differences for N20o–p, N20p–P25p, and N30o–p (Table 2, Figure 2 and Figure 3B). The amplitude ratios of N20o–p and N20p–P25p in the three patients with PMA were not different with the control values, but the ratio for N30o–p was significantly lower (more inhibited) than the control values (Table 2).

In control subjects, there was a significant negative correlation between baseline N20p–P25p and N30o–p amplitude and the gating ratio (*p* = 0.0028 and *p* = 0.0054, respectively; Pearson’s correlation efficient; Figure 4B,C), but no significant correlation between baseline N20o–p amplitude and the gating ratio (*p* = 0.0660; Figure 4A). In patients with ALS, there was a significantly negative correlation between the baseline N20o–p and N20p–P25p amplitude and the gating ratio (*p* = 0.0040 and *p* = 0.0019, respectively: Figure 4A,B), but no significant correlation between baseline N30o–p amplitude and the gating ratio (*p* = 0.1081; Figure 4C).

## 4. Discussion

The enlargement of N20 components (N20o–p and N20p–P25p) of SEPs in our patients was consistent with previous studies [4]. We found, for the first time, that frontal N30 was also enlarged in patients with ALS. Additionally, the suppression of SEP amplitude during voluntary finger movements, including for N30, was normal in patients with ALS, indicating that the sensory gating system was well preserved in these patients. Baseline amplitudes tended to increase with the gating ratio in patients and controls, indicating that the increased amplitudes in patients were not attributed to dysfunction of sensory gating.

The N20 and P25 of median nerve SEPs reflect the postsynaptic excitatory potentials of the pyramidal neurons in the primary sensory cortex [21]. The hyperexcitability of the sensory cortex in ALS, shown as the increased N20p–P25p amplitude, may be caused by neuroplastic changes in the cortical sensorimotor network [4]. The motor and sensory cortices modulate each other [22,23], and the motor cortex hyperexcitability may induce the sensory cortex hyperexcitability. Another possibility is that the neuropathological changes in the sensory cortex in ALS altered neuronal excitability. TDP-43 aggregates have been found in the sensory cortex of patients with ALS [1,2]. Our previous studies showed the N20p–P25p amplitude was increased in early stage ALS and disappeared in advanced-stage ALS [6,24]. These changes of cortical SEP components may be a result of the progressive neurodegeneration of the sensory cortex.

The frontal N30 component may originate in the secondary motor cortices, including the premotor cortex and supplementary motor cortex [19,25,26], and the phase-locking mechanism of the EEG oscillation in the beta/gamma range is suggested to contribute to frontal N30 generation [27,28]. This component has not been investigated in ALS before, although the secondary motor cortices are one of the targets of TDP-43 aggregation and subsequent neurodegeneration [1,2]. The increase in N30 amplitude in this study could be attributed to an early neurodegenerative process in ALS as well as the increase in N20 and P25 amplitude.

The neural mechanisms that produce sensory gating involve the corticofugal (top-down) and corticopetal (bottom-up) inhibitory functions to the central sensory mediating neurons, including the cortical neurons of primary sensory, primary, and secondary motor cortices, and spinal cord neurons (dorsal horn neurons) [11]. Descending or local inhibitory neurons may play a role in sensory gating during voluntary movements. If these inhibitory neurons are impaired in the neurodegenerative process of ALS, the sensory gating would also be abnormal in ALS. Our findings indicate that the inhibitory functions to the central sensory neurons were well preserved in patients with ALS and suggest that the increased N20 and N30 amplitude was due to primary hyperexcitability of cortical neurons. This hypothesis agrees with our previous study, in which the amplitudes of high-frequency oscillation in the N20 were well preserved in ALS [5].

The three patients with PMA also showed increased amplitudes of SEP in this study. Although patients with PMA exhibit no clinical upper motor neuron signs, this category has been considered equivalent to typical ALS and usually shows a similar rate of progression. As discussed above, if the enlargement of cortical SEP components results from sensory cortical lesions, our findings in PMA may certainly be caused by subclinical central nervous dysfunction. The reason why the amplitude reduction in N30 during movements was enhanced in patients with PMA is unknown, but this indicates that the gating function was well preserved.

This study has limitations. First, the sample size was small. However, we believe that, even from a small number of patients, a clear conclusion can be drawn from the fact that sensory gating was normal despite sensory cortical hyperexcitability in individual patients. Second, the extent of voluntary finger movement could not be controlled. In paretic hands, the effect of finger movements on sensory neurons might be lower in patients with ALS than in control subjects. Nevertheless, the results were similar between patients and controls, leading to the conclusion that sensory gating is preserved in ALS. Conversely, sensory gating may be enhanced in ALS since more effort is required to move the fingers for patients with paretic hands, potentially inducing hyperactivity in corticofugal inhibitory pathways. This idea may be supported by the results of the three patients with PMA. Third, we did not examine gene variants associated with ALS in the patients. Patients with a Cu/Zn superoxide-dismutase-1 gene variant have been reported to have peripheral sensory neuropathy [1,3]. Although our patients showed no sensory symptoms or abnormalities of SNAP, subclinical sensory neuropathy might still have some effects on central sensory function. Fourth, we have not yet investigated the relationship between sensory gating and disease progression. A follow-up study is needed to elucidate the chronological variations in central sensory regulation in a larger group of patients with ALS.

In conclusion, our results suggest that sensory gating function is preserved in ALS, at least in the early stage. Therefore, the enhanced median nerve SEPs may be attributed to the hyperexcitability of the primary sensory cortex. Preserving sensory gating function in early stage ALS may hold potential clinical benefits. If adequate motor rehabilitation can reduce sensory cortical excitability, it may induce a temporary recovery of cortical sensorimotor network functions, although there is no evidence that motor rehabilitation improves the survival prognosis in ALS. Further investigation is needed to elucidate sensorimotor network abnormalities and their clinical relevance in ALS.

## Figures and Tables

**Figure 1 brainsci-13-01325-f001:**
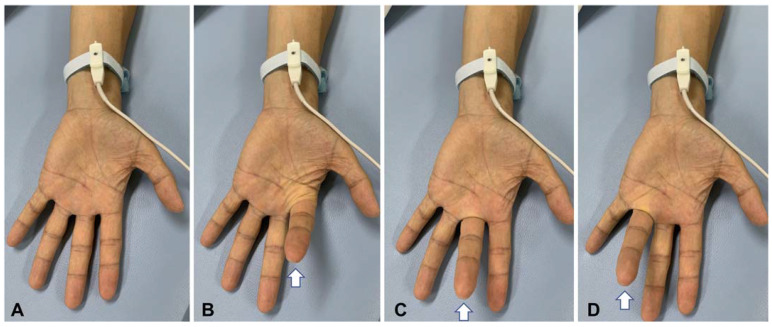
Photographs show median nerve stimulation at baseline (rest, (**A**)) and during a task (**B**–**D**). Participants moved their fingers of the stimulated hand alternatively to produce the act of “playing the piano” at an approximate rate of one to two movements per second. Participants were asked to move their fingers in such a way as to avoid rubbing the fingers against each other (see text). The hand shown in the figure is that of a control subject. Arrows indicate fingers with a voluntary flexion movement.

**Figure 2 brainsci-13-01325-f002:**
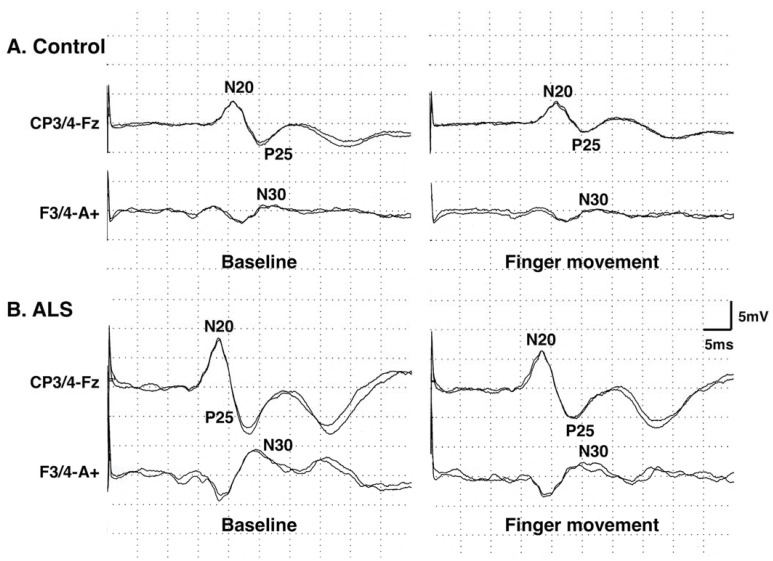
Representative waveforms of the median nerve somatosensory evoked potentials (SEPs) in a control subject (**A**) and a patient with amyotrophic lateral sclerosis (ALS) (**B**), recorded at rest (baseline) and during a task (finger movement). A+, F3/4, Fz, and CP3/4 represent the positions of electrodes on the scalp.

**Figure 3 brainsci-13-01325-f003:**
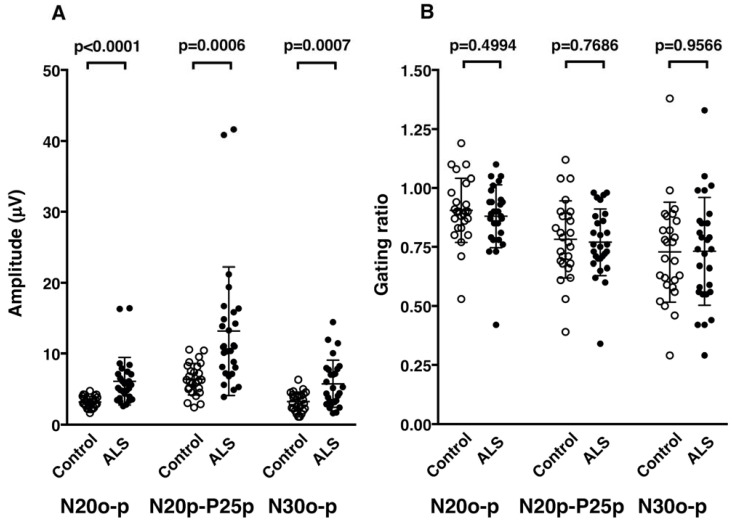
(**A**) Amplitude of each component of somatosensory evoked potentials (SEPs) in control subjects (open circles) and patients with amyotrophic lateral sclerosis (ALS) (closed circles). (**B**) The gating ratio of each SEP component calculated as the ratio of the amplitude during finger movement relative to the amplitude at rest in controls (open circles) and patients with ALS (closed circles). The dots in the figure represent the value for each median nerve stimulation; one participant had two data points from bilateral median nerve stimulation.

**Figure 4 brainsci-13-01325-f004:**
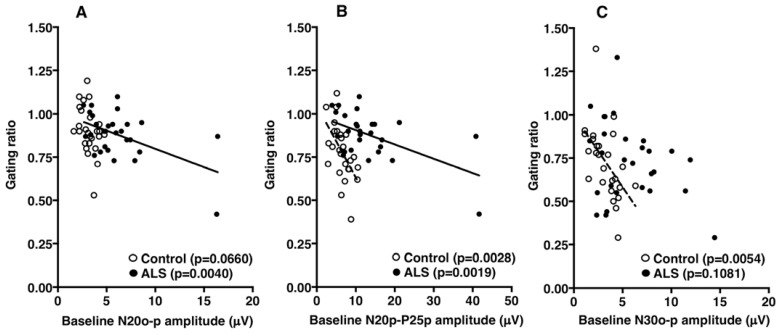
(**A**) Correlations between the baseline amplitude of the N20 onset–peak (N20o–p) and the gating ratio in controls (open circles) and patients with amyotrophic lateral sclerosis (ALS) (closed circles). (**B**) Correlations between the baseline N20 peak–P25 peak (N20p–P25p) amplitude and the gating ratio. (**C**) Correlations between the baseline amplitude of N30 onset–peak (N30o–p) and the gating ratio. Solid and dotted lines indicate significant linear regressions in patients and controls, respectively.

**Table 1 brainsci-13-01325-t001:** Characteristics of the study participants.

	Patients	Controls	*p*-Value
Number	14	13	
Sex (male:female)	9:5	7:6	
Age at onset (years)	65.4 ± 8.9	–	
Age at examination (years)	67.2 ± 8.4	66.2 ± 6.1	0.7631
Onset region (bulbar:spinal)	3:11	–	
Disease duration (years)	1.8 ± 1.0	–	
ALSFRS-R	35.8 ± 6.0	–	
Body mass index (kg/m^2^)	19.6 ± 3.9	–	
Forced vital capacity (%)	67.7 ± 22.3	–	
CMAP amplitude (mV) *	3.0 ± 3.3	7.3 ± 1.6	<0.0001
SNAP amplitude (µV) *	27.6 ± 12.6	26.0 ± 9.7	0.2643

ALSFRS-R, revised Amyotrophic Lateral Sclerosis Functional Rating Scale. CMAP, compound muscle action potential of the median nerve. SNAP, sensory nerve action potential of the median nerve. * CMAP was recorded from the abductor pollicis brevis muscle. SNAP was recorded from the index finger. Averages of CMAP and SNAP amplitude were calculated for unilateral median nerves of the weaker hand of patients with amyotrophic lateral sclerosis and the left hand of control subjects. *p*-value was obtained using Welch’s *t*-test.

**Table 2 brainsci-13-01325-t002:** Results of somatosensory evoked potentials and sensory gating.

	All Patients	Controls	*p*-Value	PMA Patients	*p*-Value
Amplitude at rest (µV)					
N20o–p	6.12 ± 0.63	3.19 ± 0.16	<0.0001	9.22 ± 5.61	0.0464
N20p–P25p	13.18 ± 1.71	6.41 ± 0.43	0.0006	22.90 ± 14.37	0.0373
N30o–p	5.76 ± 0.63	3.25 ± 0.26	0.0007	8.33 ± 4.26	0.0327
Amplitude ratio					
N20o–p	0.88 ± 0.14	0.90 ± 0.14	0.4994	0.82 ± 0.20	0.3482
N20p–P25p	0.77 ± 0.14	0.78 ± 0.16	0.7686	0.68 ± 0.18	0.2358
N30o–p	0.73 ± 0.23	0.73 ± 0.21	0.9566	0.54 ± 0.16	0.0362

PMA, progressive muscular atrophy.

## Data Availability

The raw data used in this study are not publicly available due to privacy restrictions.

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
