# Peer review of "Sensory Gating during Voluntary Finger Movement in Amyotrophic Lateral Sclerosis with Sensory Cortex Hyperexcitability"

_brainsci, 2023, doi:10.3390/brainsci13091325_

Round 1

Reviewer 1 Report

This study describes sensory findings in ALS through the use of SEPS. The authors state that the hyperexcitability of motor cortices in ALS might influence the sensory cortex’s excitability while preserving sensory gating. I find the study interesting and innovative but I have some considerations and comments:

- “ALS is multisystem neuronal degeneration including the central sensory pathways and sensory cortices”. References 1-3 are quite old. I suggest reading and citing a very recent systematic review on sensory involvement in ALS (Sensory neuropathy in amyotrophic lateral sclerosis: a systematic review. J Neurol. 2023 Aug 23. doi: 10.1007/s00415-023-11954-1). This review also updated studies on SEPS in ALS. Indeed, sensory involvement in ALS is debated as cortical dysfunction might be present but neuropathy coexists in many cases.

-“median nerve at the wrist in the more paretic hand in patients and in the left hand in control subjects.”. I was wondering why the author has made this choice (left side in controls). It would be more appropriate a stimulate of half the patients on the right and half on the left. Why the left side? Was it a way to study the non-dominant hemisphere? I think that it might be a bias and should be discussed more in detail.

-Normal SNAPs in ALS patients support central sensory dysfunction. However, the comparison between right and left SNAPs from the median nerve is not the same due to carpal tunnel syndrome (more frequent in dominant hands).

-“PMA shares a common pathophysiology with ALS ”. The clinical phenotype of PMA in ALS is quite discussed in the literature. However, it is well-known that patients with SOD1 mutations may display sensory-motor axonal damage, mimicking sensory-motor axonal polyneuropathy. This point has to be considered because the new insights provided by genetics are useful in interpreting the pathophysiology of ALS.

-“increased in early-stage ALS and disappeared in advanced-stage ALS”. A follow-up study is in demand to confirm this hypothesis. I suggest discussing it in the limitations.

-the quality of the figures is poor. Probably the figures were distorted in the process of assembling the manuscript.  

Author Response

Reply to Reviewer #1

We thank the reviewer for the positive and helpful comments. We revised the manuscript accordingly and indicated the revised points by red colored letters in the text.

  1. “ALS is multisystem neuronal degeneration including the central sensory pathways and sensory cortices”. References 1-3 are quite old. I suggest reading and citing a very recent systematic review on sensory involvement in ALS (Sensory neuropathy in amyotrophic lateral sclerosis: a systematic review. J Neurol. 2023 Aug 23. doi: 10.1007/s00415-023-11954-1). This review also updated studies on SEPS in ALS. Indeed, sensory involvement in ALS is debated as cortical dysfunction might be present but neuropathy coexists in many cases.

Our response: We thank the reviewer for the helpful comments. We deleted two references (Ref. 2 and 3 in the previous version). Instead, we cited the reference by Bombaci et al. (J Neurol, 2023) and also another review reference by Rubio et al. (Int J Mol Sci, 2022). We revised the Introduction section as follows: “Accumulated evidence, however, indicates that ALS is multisystem neuronal degeneration including the peripheral and central sensory pathways [1-3].”

  1. “median nerve at the wrist in the more paretic hand in patients and in the left hand in control subjects.”. I was wondering why the author has made this choice (left side in controls). It would be more appropriate a stimulate of half the patients on the right and half on the left. Why the left side? Was it a way to study the non-dominant hemisphere? I think that it might be a bias and should be discussed more in detail.

Our response: We examined the left median nerve (non-dominant hand) for peripheral nerve conduction study, but, for SEPs, we examined bilateral medina nerve. The nerve conduction study was performed to rule out the possibility of systemic sensory neuropathy. For SEPs, we examined both hemispheres, so we believe that there was no bias on the results of SEPs and gating ratio.

  1. Normal SNAPs in ALS patients support central sensory dysfunction. However, the comparison between right and left SNAPs from the median nerve is not the same due to carpal tunnel syndrome (more frequent in dominant hands).

Our response: We thank the reviewer for the important comments. Unfortunately, for SNAP, we only checked unilateral median nerve. In this study, however, we did not observe the findings suggesting carpal tunnel syndrome such as a delayed latency of SNAP in our patients. As mentioned above, SEPs were examined for both hemispheres, and we believe that there was no difference in the effect of laterality between patients and control subjects.

  1. “PMA shares a common pathophysiology with ALS ”. The clinical phenotype of PMA in ALS is quite discussed in the literature. However, it is well-known that patients with SOD1 mutations may display sensory-motor axonal damage, mimicking sensory-motor axonal polyneuropathy. This point has to be considered because the new insights provided by genetics are useful in interpreting the pathophysiology of ALS.

Our response: We thank the reviewer for the helpful comments. The patients in this study were all sporadic although we did not examine SOD1 gene variants. Furthermore, no patients exhibited sensory symptoms or SNAP abnormalities suggesting concomitant sensory axonal polyneuropathy. We added one sentence in the participant section as follows: “We did not perform ALS-related gene analyses for all patients in this study.” We also added the discussion in the limitation section as follows: “Third, we did not examine gene variants associated with ALS in the patients. Patients with a Cu/Zn superoxide-dismutase-1 gene variant have been reported to exhibit peripheral sensory neuropathy [1, 4]. Although our patients showed no sensory symptoms or abnormalities in the median nerve SNAP, subclinical sensory neuropathy might still have some effects on central sensory function.”

  1. “increased in early-stage ALS and disappeared in advanced-stage ALS”. A follow-up study is in demand to confirm this hypothesis. I suggest discussing it in the limitations.

Our response: We added one sentence in the limitation section as follows: “Fourth, we have not yet investigated the relationship between sensory gating and disease progression. A follow-up study is needed to elucidate chronological variations of central sensory regulation in a larger group of patients with ALS.”

  1. the quality of the figures is poor. Probably the figures were distorted in the process of assembling the manuscript.

Our response: We have revised all the figures with a format of JPEG. We hope they will be fine for publishing.

Reviewer 2 Report

Indeed, ALS is an urgent problem of modern neurology. Despite the small prevalence of this disease, the imminent disability and death of patients makes the disease interesting for scientists and practitioners. The research conducted by the authors is undoubtedly of scientific interest. However, during the review process a number of comments emerged:

1) Figures 3 and 4, are of poor quality, authors should improve the quality of drawings.

2) Table 1 provides a detailed description of the study groups, but the authors should provide another table of the results of the study.

3) The authors describe in detail the procedure of research in the text of the manuscript, it would be possible to visualize a schematic drawing or a photograph of the progress of research.

4) In the "discussion" section authors should indicate the value of their findings for practical health care, how the patterns found by authors in the future can help patients with ALS. This may be part of a new rehabilitation and/or treatment programme for these patients. This data will enable the readers of the manuscript to better understand and appreciate the value of the results obtained by the authors of the manuscript.

Author Response

(The authors gave the same response as above.)

Round 2

Reviewer 1 Report

The authors have addressed all my concerns. I have no further comments.